# The effects of plantarflexor weakness and reduced tendon stiffness with aging on gait stability

Ross E. Smith[1], Andrew D. Shelton[1], Gregory S. Sawicki[2,3], Jason R. Franz[1]*

1 Joint Dept. of Biomedical Engineering, UNC Chapel Hill and NC State University, Chapel Hill, North Carolina, United States of America, 2 George W. Woodruff School of Mechanical Engineering, Georgia Institute of Technology, Atlanta, Georgia, United States of America, 3 School of Biological Sciences, Georgia Institute of Technology, Atlanta, Georgia, United States of America

* jrfranz@email.unc.edu

**Data Availability Statement:** A .mat master data file is available here: https://cdr.lib.unc.edu/concern/data_sets/6d5707327.

**Funding:** This project was supported by NIH Grants R01AG058615 to JRF and GSS and

## Abstract

Falls among older adults are a costly public health concern. Such falls can be precipitated by balance disturbances, after which a recovery strategy requiring rapid, high force outputs is necessary. Sarcopenia among older adults likely diminishes their ability to produce the forces necessary to arrest gait instability. Age-related changes to tendon stiffness may also delay muscle stretch and afferent feedback and decrease force transmission, worsening fall outcomes. However, the association between muscle strength, tendon stiffness, and gait instability is not well established. Given the ankle's proximity to the onset of many walking balance disturbances, we examined the relation between both plantarflexor strength and Achilles tendon stiffness with walking-related instability during perturbed gait in older and younger adults–the latter quantified herein using margins of stability and whole-body angular momentum including the application of treadmill-induced slip perturbations. Older and younger adults did not differ in plantarflexor strength, but Achilles tendon stiffness was lower in older adults. Among older adults, plantarflexor weakness associated with greater whole-body angular momentum following treadmill-induced slip perturbations. Weaker older adults also appeared to walk and recover from treadmill-induced slip perturbations with more caution. This study highlights the role of plantarflexor strength and Achilles tendon stiffness in regulating lateral gait stability in older adults, which may be targets for training protocols seeking to minimize fall risk and injury severity.

## 1. Introduction

Up to 35% of older adults fall annually, resulting in serious injuries and significant healthcare costs. Moreover, the likelihood of incurring severe injuries from a fall increase with age [1]. The nature of age-related falls risk is likely multifactorial, and the literature has catalogued numerous environmental and intrinsic factors that may precipitate or, with proper intervention, help prevent falls from occurring [2–5]. Unfortunately, despite these efforts, these concerns are accelerating; in the last 10 years, the death rate from falls has increased nearly 30%

R21AG067388 to JRF. National Institutes of Health (nih.gov). These sponsors played no role in study design, data collection or analysis, decision to publish, or preparation of the manuscript.

**Competing interests:** The authors have declared no competing interests exist.

and fall-related hospitalizations among older adults has increased more than 200% [6]. Thus, there remains a critical need to identify novel, evidence-based, and modifiable factors to increase older adults' resilience against balance challenges to mitigate the severity of fall-related injuries.

Balance perturbations precipitating falls are most commonly experienced during walking, where mitigation of instability requires rapid repositioning of the base of support and sufficient ground reaction forces from the musculoskeletal system to arrest momentum. In younger adults, dynamic balance recovery following gait perturbations are partly dependent on large, rapid leg extensor moments [7, 8]. With advanced age, adults experience sarcopenia [9, 10] and decreased muscle strength [11, 12], which would intuitively hinder balance recovery in older adults. Indeed, older adults with reduced maximal isometric leg press strength are more likely to fall [13]. Also, older adults are known to have age-related reductions in series elastic tissue stiffness [14–17]. In addition to disrupting metabolic walking cost and plantar-flexor muscle neuromechanics [18], decreased tissue stiffness is associated with reducing maximal joint torque by increasing muscle shortening velocities [19–21] and increasing electromechanical delay [22], both of which likely impose further penalties to balance recovery in older adults.

The relationship between age-related declines in muscle-tendon unit (MTU) integrity and dynamic stability has been explored during a variety of tasks. Due to the nature of common, real-world balance perturbations (e.g. tripping, slipping) and the ankle's proximity to the onset of such perturbations, the triceps surae and Achilles tendon are often the locus of investigation for this relationship. Onambele et al. [16] found older adults were more unstable, had weaker ankle plantarflexors and lesser Achilles tendon stiffness ($k_{AT}$) compared to younger adults showed during single-leg stance and tandem stance. During simulated forward falling, older adults took more steps to recover than younger adults, and had weaker knee and ankle extensors, as well as less stiff patellar tendons with no difference in $k_{AT}$ [23, 24]. While neither of these experimental tasks involve gait, both indicate age-related declines in MTU integrity are important to postural stability. Considering gait stability in response to perturbations, Epro et al. [25] showed older adult women with stronger ankle plantarflexors and greater $k_{AT}$ had better gait stability after unexpected treadmill belt accelerations. However, they later found increases in MTU capacity at the ankle through exercise had minimal effects on balance recovery in a similar population [26]. Most recently, Debelle et al. [8] showed that age-related declines in MTU integrity were not related to balance recovery, as older and younger adults showed similar margins of stability in the anteroposterior (AP) direction. This finding is unusual, as older adults are often reported to be more unstable in the AP direction than younger adults immediately following AP balance perturbations [24, 27, 28], which the authors attribute to a comparatively young "older adult" sample. Furthermore, recent literature demonstrated that AP gait perturbations were more closely correlated with compensatory efforts in the mediolateral (ML) direction during balance recovery [29]. Given that ML balance is disproportionately worsened with age [30, 31], related to increased falls risk [32], and is known to require greater active sensorimotor control from leg muscles [33], quantification of the relationship between MTU integrity and both AP and ML stability may provide novel insight into why older adults are more vulnerable to falls following gait perturbations.

Overall, the evidence concerning age-related declines in MTU integrity and their effect on older adults' gait stability is not clear, as older adult groups have inconsistently reported age-range cutoffs; the existing literature differ among perturbation paradigms (e.g. forward falling, mechanical pulling or "trip", and slip); of these studies concerned with gait perturbations, none report metrics concerning mediolateral stability in either young or older adults; and Achilles and patellar tendon stiffness are inconsistently different between older and younger

adults when correlated with any of the various stability metrics. Therefore, our purpose was to determine whether ankle plantarflexor muscle strength and $k_{AT}$ associated with vulnerability to walking treadmill-induced slip perturbations in both young and older adults, which is summarized in Fig 1. We first hypothesized that older adults would exhibit reduced plantarflexor strength and lesser $k_{AT}$. Additionally, reductions in plantarflexor strength and $k_{AT}$. would be correlated with larger perturbation-induced changes due to treadmill-induced slips when compared to younger adults. Ultimately, data in support of these associations would implicate age-related reductions in ankle plantarflexor muscle strength and $k_{AT}$ as local MTU mechanisms, revealing potentially modifiable factors underlying the instability elicited by slips or the severity of falling-related injuries.

## 2. Methods

### 2.1 Participants

22 healthy younger adults (10 female, age: 21.7 ± 2.0 yrs, height: 1.72 ± 0.08 m, mass: 66.4 ± 8.6 kg) and 21 healthy older adults (13 female, age: 74.0 ± 6.0 yrs, age range: 65–87, height: 1.69 ± 0.11 m, mass: 69.5 ± 18.6 kg) participated. Participants were included that had no history of neurological diseases, lower limb injury within the last 6 months, and who could walk without an assistive device. All experimental procedures and recruitment procedures were approved by the University of North Carolina at Chapel Hill Institutional Review Board (20–0555). Participant recruitment began on 4/14/20 and continued until 12/31/22. All participants gave their written informed consent prior to participation in the experimental protocols.

### 2.2 Experimental procedures

The laboratory visit began by recording participants' preferred walking speed using photocells (Bower Timing Systems, Draper, Utah, USA) from the average time of four 30-meter walking trials. Participants then completed a 3-minute warm-up walking trial on a dual-belt, instrumented treadmill (Bertec, Columbus, Ohio, USA). Participants then completed two treadmill walking trials–namely, 2 minutes of usual, unperturbed walking and walking while responding to treadmill-induced slip perturbations of a duration (200 ms) and acceleration (6 m/s$^2$)

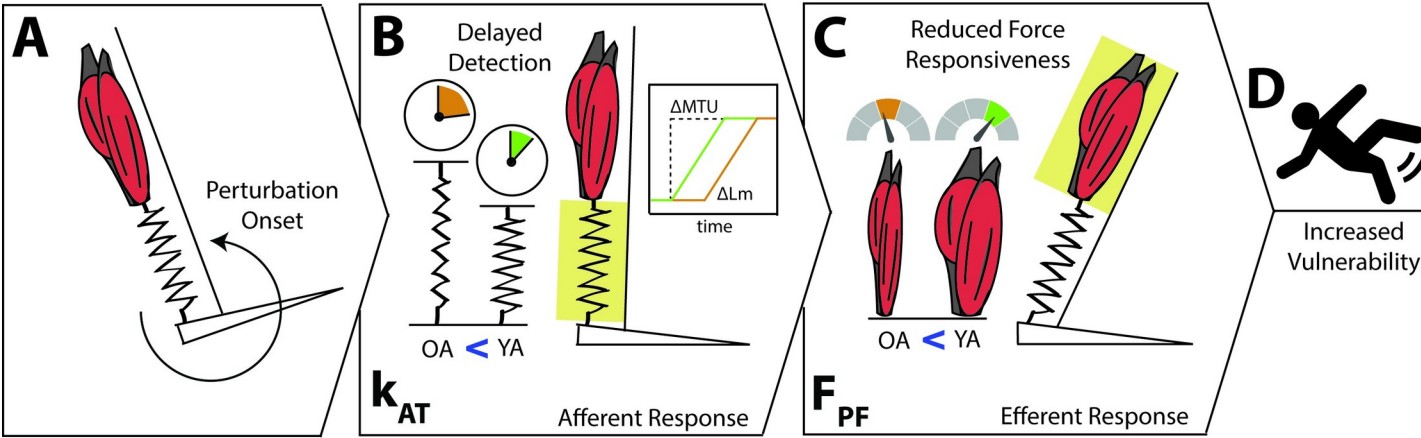

**Fig 1. Conceptual model for how age-related differences in ankle muscle-tendon unit function can increase the vulnerability to unexpected balance disturbances.**
A) The onset of a balance perturbation elicits a rapid change in joint position and thus muscle-tendon unit (MTU) lengthening. B) Lesser Achilles tendon stiffness (kAT) in older adults delays the transmission between ΔMTU length and muscle stretch (i.e., ΔLm), thereby delaying afferent detection of the perturbation compared to younger adults. C) Lesser plantarflexor muscle strength (FPF) in older adults reduces force responsiveness during perturbation recovery. D) Age-related decreases in kAT and FPF predispose older adults to increased vulnerability to unexpected balance disturbances.

shown to elicit walking-related instability. The perturbations were applied randomly at 5 heel strikes on each leg, each separated by at least 10 steps [34–36]. For all walking trials after the warm-up, a 14-camera motion capture system (Motion Analysis Corporation, Rohnert Park, California, USA) operating at 100 Hz captured the 3D positions of retroreflective markers placed on the anterior and posterior iliac spines, sacrum, lateral femoral condyles, lateral malleoli, posterior calcanei, first and fifth metatarsal heads, acromial, 7th cervical spine, 10th thoracic spine, sternum, and the sternal notch plus an additional 14 tracking markers placed in clusters on the lateral thighs and shanks.

At the conclusion of the visit, we used in vivo ultrasound imaging to characterize plantarflexor strength and $k_{AT}$. For the former, participants performed two maximal voluntary isometric plantarflexion contractions ($MVIC_{PF}$) on a dynamometer (Biodex System 4 Pro, Shirley, New York, USA) with the ankle at 90˚ (neutral) and the knee flexed to 20˚. All participants received identical verbal encouragement to give maximal effort (i.e. "push the ball of your foot like a gas pedal as hard as possible using only your calf muscles"), and adequate rest time was given between two repetitions to avoid fatigue. For the latter, we recorded the position of participants' distal medial gastrocnemius musculotendinous junction (MTJ) with the Achilles tendon (AT) using B-mode ultrasound images while participants performed two passive, isokinetic ankle rotations (20˚/s) from 20˚ plantarflexion to 30˚ dorsiflexion on the same dynamometer. The ultrasound system operated at 61 fps at a depth of 50 mm using a 60 mm linear array transducer (LV7.5/60/128Z-2, UAB Telemed, Vilnius, Lithuania) secured to the posterior shank using a custom probe mount and self-adhesive bandaging. Even bandage pressure over the probe was applied to so as not to alter muscle geometry within the image [37]. We collected motion capture from a posterior calcaneal marker as well as a marker triad rigidly affixed to the ultrasound transducer to estimate AT length change (detailed in Section 2.3.2).

## 2.3 Data processing and analysis

**2.3.1 Kinematic data.**   All marker data were filtered using a 4th order low-pass Butterworth filter with a cutoff frequency of 12 Hz. We also included a static standing calibration and hip circumduction task to establish segmental dimensions, knee and ankle joint centers, and functional hip joint centers.

**2.3.2 Plantarflexor strength and Achilles tendon stiffness.**   We calculated plantarflexor strength from the greater of the two peak ankle moment values per participant measured during $MVIC_{PF}$ trials. All $MVIC_{PF}$ values are normalized to body mass. For $k_{AT}$, two investigators manually tracked the MTJ positions for both age groups, which were divided evenly between them to ensure scientific rigor (Tracker, V 6.0.10). The trackers selected the most distal location of the gastrocnemius muscle before intersection with the AT. Based on published experience [38], the positional change of MTJ during isokinetic ankle rotations is relatively linear. Thus we manually identified the orientation of the distal deep and superficial aponeuroses every fifth frame, and their intersection defined the MTJ position, then interpolated MTJ position over the range of ankle joint rotation. The tracked 2D MTJ position data were transformed into a common coordinate system using marker data from the ultrasound probe. The distance between the transformed 3D MTJ position and the posterior calcaneus marker (i.e., a surrogate for AT insertion) was used to estimate AT length. We calculated AT force by dividing the measured ankle moment by a generalized AT moment arm length from previously published literature [39]. We assumed constant values for Achilles tendon moment arms, though we acknowledge that these measures can vary with joint position and muscle loading [40]. However, our previously published data suggest that those variations are

disproportionately small and insensitive to the effects of muscle loading in dorsiflexion. We opted to analyze only dorsiflexion ranges of motion for our estimates of Achilles tendon stiffness not only to avoid errors from the tendon's dorsoventral curvature, but also from interindividual variation in resting length. Finally, we calculated $k_{AT}$ as the change in AT force divided by the change in AT length between 20 and 80% of passive dorsiflexion rotation range. For all participants, $k_{AT}$ values were averaged between the two passive dorsiflexion trials.

**2.3.3 Balance outcomes.** We included two outcomes that quantify fundamentally-different content related to walking stability–namely, whole-body angular momentum (WBAM) and margin of stability (MoS). WBAM quantifies the summed maintenance and control of the momentum of individual body segments. Conversely, MoS quantifies a biophysical relation between the body's center of mass (CoM) and the base of support. Both are critical features of walking balance and vulnerability to treadmill-induced slip perturbations.

To calculate WBAM, we used participant-specific computational simulations implemented in OpenSim to determine segmental center of mass (CoM) positions and velocities. WBAM was calculated using previously published methods [41]. We normalized WBAM values to participant body mass (kg), height (m), and walking speed (m/s). Values are reported as a percentage of the gait cycle and averaged across the unperturbed walking trial in the frontal, transverse, and sagittal planes. We extracted the range from minimum to maximum WBAM values during unperturbed and perturbed walking across the gait cycle in all planes (i.e. $WBAM_{SagRange}$, $WBAM_{FrontRange}$, $WBAM_{TransRange}$), as well as perturbation-induced effects as the difference from unperturbed walking in all planes (i.e. $\Delta WBAM_{Sag}$, $\Delta WBAM_{Front}$, $\Delta WBAM_{Trans}$).

As a complement to WBAM, we also calculated MoS using previously published methods [42, 43]. Briefly, ML and AP MoS ($MoS_{Lat}$, $MoS_{Ant}$) were calculated as the lateral and anterior distance, respectively, between the BoS and the extrapolated CoM projected to the treadmill belt at heel strike. We defined the lateral and anterior margins of the base of support as the fifth and first metatarsal marker, respectively. For treadmill induced slips, the heel strike of interest was that of the recovery step after the application of the perturbation, whereas any heel strike was used for unperturbed walking. We also calculated perturbation-induced effects as the percent change from unperturbed walking (i.e., $\Delta MoS_{Lat}$, $\Delta MoS_{Ant}$) as a way to normalize MoS within subjects.

## 2.4 Statistics

Independent-samples t-tests assessed the effects of age on plantarflexor strength and $k_{AT}$. We then performed a series of multivariate linear regression analyses (SPSS V28, Chicago, Illinois, USA) to assess whether plantarflexor strength (i.e., peak ankle moment) and $k_{AT}$ associated with MoS and WBAM during perturbed walking or their respective perturbation-induced changes (i.e., $\Delta MoS$). We established statistical significance for all statistical tests at $p \leq 0.05$. All results are reported as mean values ± standard deviation.

## 3. Results

Fig 2 summarizes plantarflexor strength and $k_{AT}$ in younger and older adults. Older adults (0.75 ± 0.38 Nm/kg) exhibited lesser but statistically indistinguishable values of plantarflexor strength compared to younger adults (0.78 ± 0.34 Nm/kg) (p = 0.791). Conversely, we found that older adults averaged ~30% lesser $k_{AT}$ than younger adults during passive rotation (4.33 ±1.78 N/mm vs. 6.12±2.82 N/mm, p = 0.016). Plantarflexor strength and $k_{AT}$ were not correlated (see S1 Appendix).

Table 1 summarizes all individual associations between plantarflexor strength, $k_{AT}$, and balance outcomes, whereas Figs 3–6 highlights those associations and statistically significant

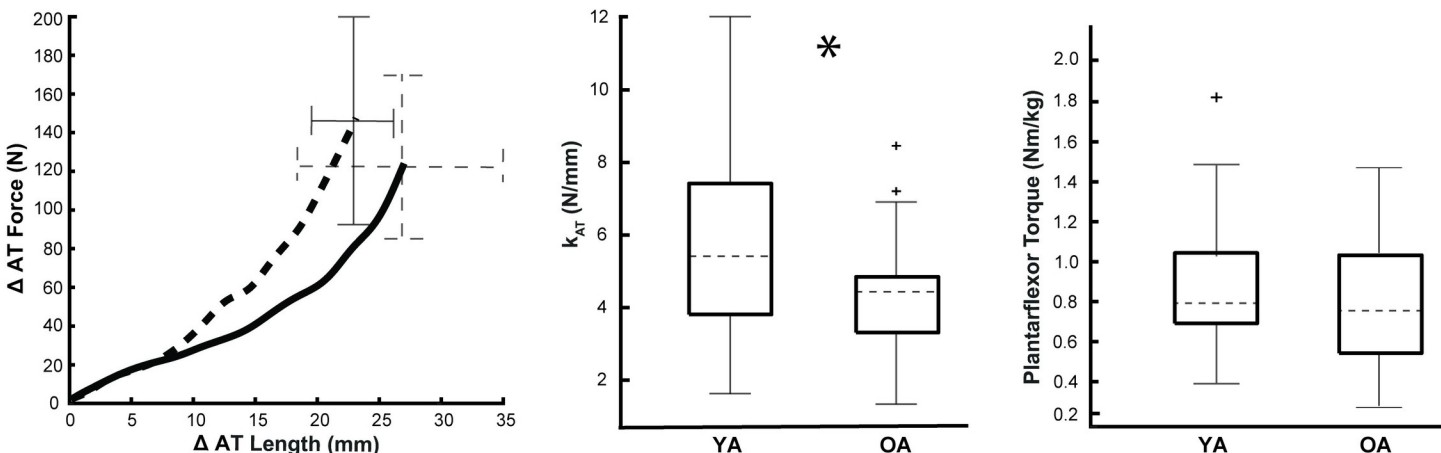

**Fig 2. Plantarflexor strength and Achilles tendon stiffness for older and younger adults.** Group mean data for older (OA, solid line, $n = 21$) and younger (YA, dashed line, $n = 22$) adults' Achilles tendon (AT) force and length changes during passive dorsiflexion (left), AT stiffness ($k_{AT}$) (middle), and peak plantarflexor torque obtained from maximal-voluntary plantarflexion contractions (right). Statistical significance is denoted by asterisk(s) and statistical outliers are denoted by plus signs. ($p = 0.05$).

results. We found no correlation between our predictive variables, $k_{AT}$ and plantarflexor strength (S1 Appendix), and in all cases, the collinearity threshold of various inflation factor set at 3 was not crossed. Lesser plantarflexor strength among older adults was associated with greater $MoS_{Lat}$ (Fig 3) ($r^2 = 0.194$, $F(1,19) = 3.053$, $p = 0.023$). There were no other significant correlations for any other MoS metric in any plane. Among OA, greater $WBAM_{TransRange}$ was associated with lesser plantarflexor strength and greater $k_{AT}$ (Fig 4) (Full model-$r^2 = 0.372$, $F(2,18) = 5.332$, $p = 0.015$; Plantarflexor Torque-$r^2 = 0.266$, $p = 0.008$; $k_{AT}$-$r^2 = 0.165$, $p = 0.034$). We found no significant correlations between any metric among younger adults, or for any perturbation-induced changes (Figs 5 and 6).

## 4. Discussion

Our purpose was to determine whether plantarflexor muscle strength and $k_{AT}$ associated with the effect of walking treadmill-induced slip perturbations on dynamic balance recovery across

**Table 1. Relations between Achilles tendon stiffness and plantarflexor torque and balance outcomes during perturbed walking.**

| | YA | | | | OA | | | |
|---|---|---|---|---|---|---|---|---|
| | $K_{AT}$ | | PF Torque | | $K_{AT}$ | | PF Torque | |
| | *r* | *p* | *r* | *p* | *r* | *p* | *r* | *p* |
| **Margin of Stability (MoS)** | | | | | | | | |
| $MoS_{Lat}$ | -0.339 | 0.061 | -0.077 | 0.366 | 0.313 | 0.084 | **-0.440** | **0.023** |
| $\Delta MoS_{Lat}$ | -0.356 | 0.052 | -0.063 | 0.390 | -0.249 | 0.138 | -0.019 | 0.467 |
| $MoS_{Ant}$ | -0.144 | 0.262 | -0.204 | 0.181 | 0.084 | 0.359 | -0.152 | 0.255 |
| $\Delta MoS_{Ant}$ | -0.146 | 0.258 | -0.237 | 0.144 | -0.083 | 0.360 | 0.192 | 0.203 |
| **Whole-body Angular Momentum (WBAM)** | | | | | | | | |
| $WBAM_{SagRange}$ | -0.011 | 0.480 | -0.180 | 0.212 | 0.018 | 0.469 | -0.158 | 0.247 |
| $\Delta WBAM_{Sag}$ | 0.079 | 0.363 | -0.232 | 0.150 | 0.003 | 0.495 | -0.091 | 0.347 |
| $WBAM_{FrontRange}$ | -0.049 | 0.414 | -0.203 | 0.182 | 0.155 | 0.311 | -0.347 | 0.065 |
| $\Delta WBAM_{Front}$ | 0.064 | 0.388 | -0.082 | 0.359 | -0.135 | 0.279 | -0.161 | 0.242 |
| $WBAM_{TransRange}$ | -0.001 | 0.498 | -0.213 | 0.170 | **0.406** | **0.034** | **-0.516** | **0.008** |
| $\Delta WBAM_{Trans}$ | 0.341 | 0.060 | -0.085 | 0.354 | -0.015 | 0.474 | -0.153 | 0.253 |

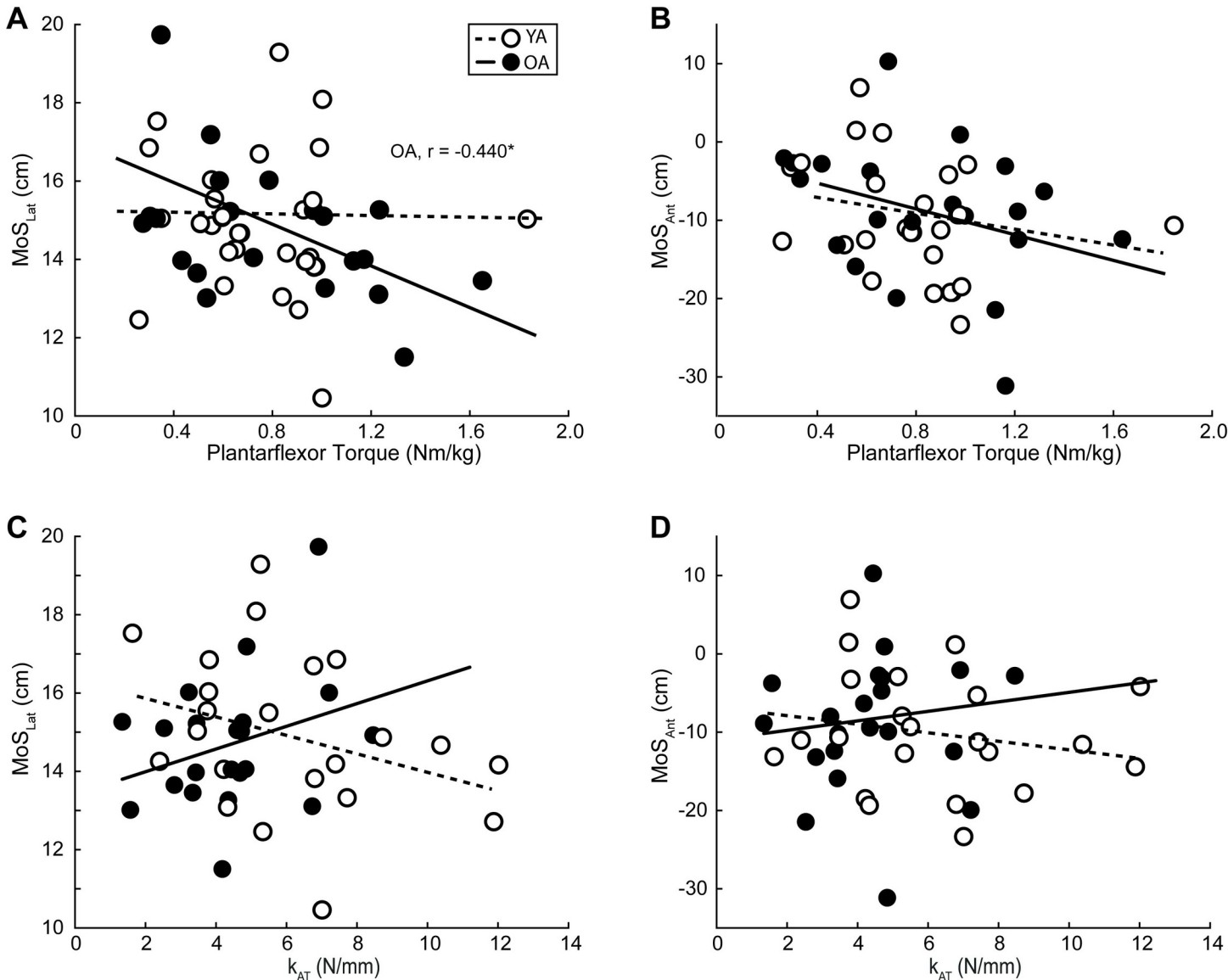

**Fig 3. Plantarflexor strength and Achilles tendon stiffness relations to margins of stability during perturbed walking.** Individual average peak plantarflexor torque, Achilles tendon stiffness ($k_{AT}$), mediolateral ($MoS_{Lat}$) and anteroposterior ($MoS_{Antt}$) direction margins of stability for older (solid line and solid dot, $n = 21$) and younger (dashed line and open dot, $n = 22$) adults during the recovery step following treadmill-induced slip perturbations. Lines of best fit for both groups are shown and significant correlations are depicted per group by their corresponding r-value accompanied by an asterisk. ($p = 0.05$).

a cohort of younger and older adults. Our results were in partial agreement with our hypotheses. First, older adults exhibited lesser $k_{AT}$ but not lesser plantarflexor strength than younger adults. Second, we revealed associations between both plantarflexor strength and $k_{AT}$ and key balance outcomes. For example, we found that weaker older adults generally exhibited greater vulnerability (i.e. larger WBAM) in the transverse plane following treadmill-induced slip perturbations. Together, these results reveal specific features of muscle-tendon units spanning the ankle that are relevant to walking balance control and which may be targeted through exercise interventions to mitigate the risk of falls among older adults.

In this study, older adults had similar plantarflexor strength as young adults during $MVIC_{PF}$ contractions. Given the well-documented nature of age-related declines in muscle

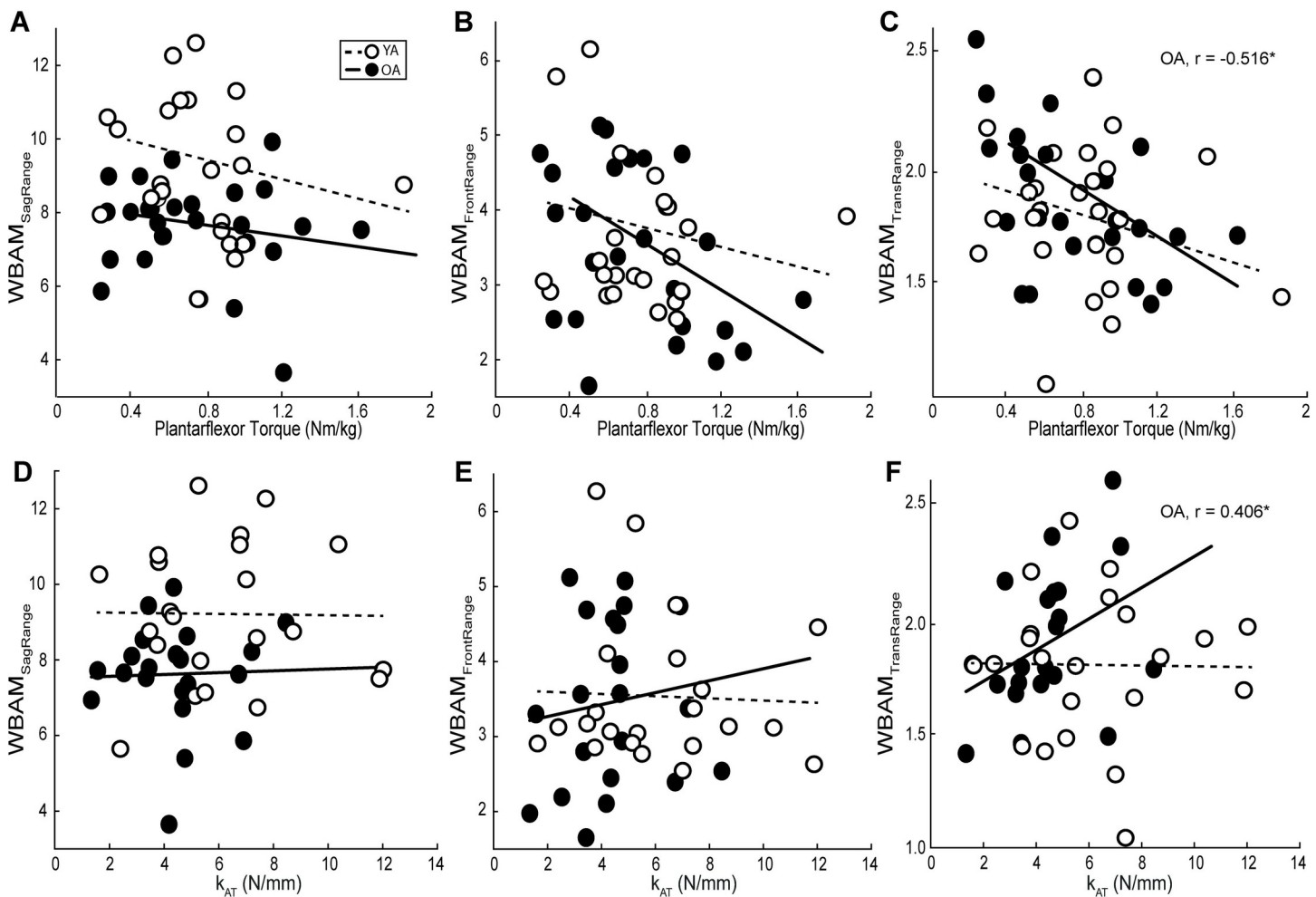

**Fig 4. Plantarflexor strength and Achilles tendon stiffness relations to whole-body angular momentum during perturbed walking.** Individual average peak plantarflexor torque, Achilles tendon stiffness ($k_{AT}$), and whole-body angular momentum in the sagittal (WBAM$_{SagRange}$), frontal (WBAM$_{FrontRange}$), and transverse planes (WBAM$_{TransRange}$) for older (solid line and solid dot, $n = 21$) and younger (dashed line and open dot, $n = 22$) adults during during the recovery step following treadmill-induced slip perturbations. Lines of best fit for both groups are shown and significant correlations are depicted per group by their corresponding r-value accompanied by an asterisk. ($p = 0.05$).

strength, similar values between age groups were surprising. We may attribute our uncharacteristically strong older adults as a potential feature of their high physical activity levels documented using a standard health screening. Another explanation may be that normalized ankle torque values among our younger adults are low when compared with previously reported values [8, 16, 24], which could result from misalignment of ankle joint rotation axes with that of the dynamometer or insufficient recruitment of ankle musculature in younger adults. As we accounted for alignment of dynamometer and ankle joint axes of rotation in our setup [44] and gave consistent vocal encouragement for maximal effort to all participants [45], it is unclear why MVIC$_{PF}$ torque values among our younger adults were low in comparison with previous values. Nevertheless, only older adults with weaker plantarflexors had perturbation-induced changes to their stability, which we first discuss for WBAM and then for MoS. Regulation of sagittal-plane WBAM is considered a critical feature of walking stability [46, 47] and balance recovery [48, 49]. Notably, a hip-dominant strategy was previously identified for controlling sagittal-plane WBAM following combined slip and upper-body perturbations acting

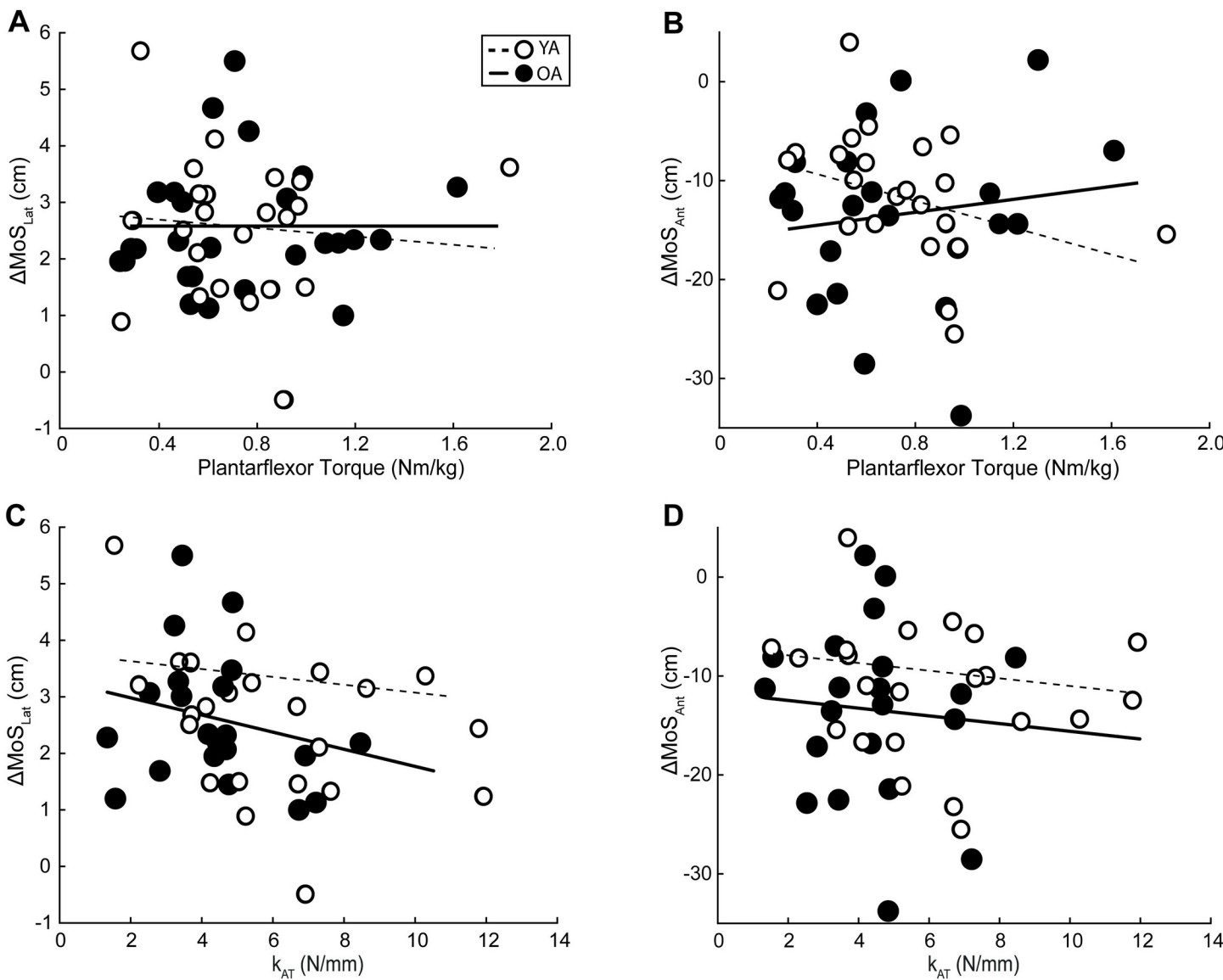

**Fig 5. Plantarflexor strength and Achilles tendon stiffness relations to relative margins of stability during perturbed walking.** Individual average peak plantarflexor torque, Achilles tendon stiffness ($k_{AT}$), relative mediolateral ($\Delta MoS_{Lat}$) and anteroposterior ($\Delta MoS_{Ant}$) direction margins of stability for older (solid line and solid dot, $n = 21$) and younger (dashed line and open dot, $n = 22$) adults during the recovery step following treadmill-induced slip perturbations. Lines of best fit for both groups are shown and significant correlations are depicted per group by their corresponding r-value accompanied by an asterisk. ($p = 0.05$).

in opposite directions for healthy young adults [49]. Sagittal-plane WBAM was similar between fallers and non-fallers in a small sample of older adults walking at their preferred speed [50]. However, we also showed that plantarflexor strength can differentiate older adults' ability to control WBAM in the transverse plane following a treadmill-induced slip perturbation. This idea is supported by findings that demonstrated the plantarflexors' contributions to controlling WBAM in outside of the sagittal plane to counteract angular momentum of the body's center of mass [51, 52]. Thus, stronger plantarflexors would provide a mechanism to mitigate increased transverse-plane WBAM following a treadmill-induced slip perturbation.

Older adults with weaker plantarflexors exhibited larger $MoS_{Lat}$ when responding to treadmill-induced slip perturbations–an outcome that would represent a more stable gait by

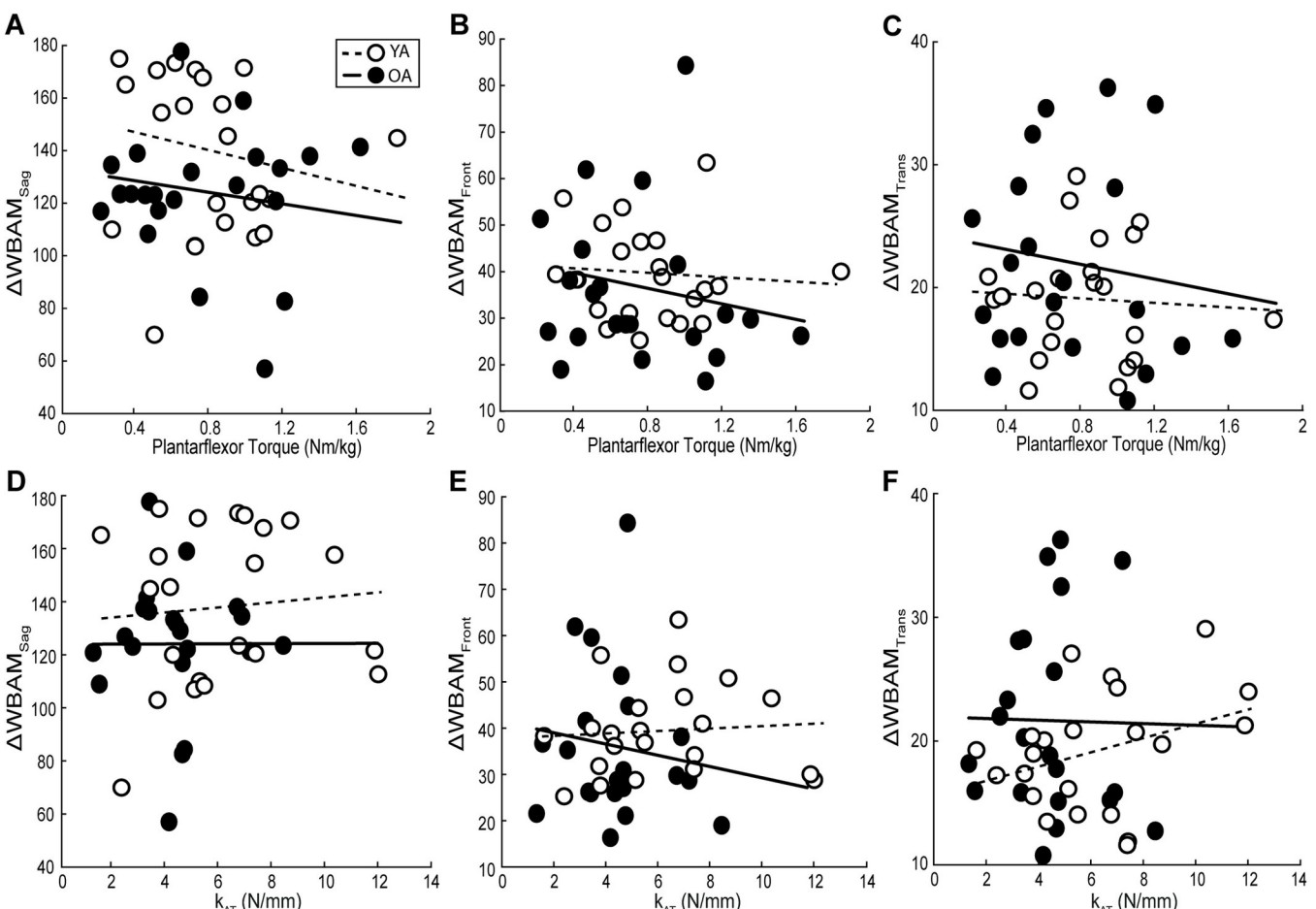

**Fig 6. Plantarflexor strength and Achilles tendon stiffness relations to relative whole-body angular momentum during perturbed walking.** Individual average peak plantarflexor torque, Achilles tendon stiffness ($k_{AT}$), and relative whole-body angular momentum in the sagittal ($\Delta WBAM_{SagRange}$), frontal ($\Delta WBAM_{FrontRange}$), and transverse planes ($\Delta WBAM_{TransRange}$) for older (solid line and solid dot, $n = 21$) and younger (dashed line and open dot, $n = 22$) adults during the recovery step following treadmill-induced slip perturbations. Lines of best fit for both groups are shown and significant correlations are depicted per group by their corresponding r-value accompanied by an asterisk. ($p = 0.05$).

common interpretations [42, 43]. Although this finding is initially counterintuitive, it is not in itself novel. Mademli and Arampatzis [53] showed that older adults walk normally with higher $MoS_{Ant}$ than younger adults. Those authors interpreted their findings as indicative of a more cautious strategy adopted by older adults. As evidenced by our results, we contend that this strategy also affects how relatively weaker adults attenuate perturbations, particularly in the ML direction. Ultimately, perhaps strength represents a measure of balance recovery confidence, where stronger older adults were more comfortable operating nearer their limits of stability when attenuating a treadmill-induced slip perturbation. This concept extends the interpretation of MoS to consider task context and habitual behaviors like walking speed when cataloging the literature on both steady-state gait behaviors and responses to walking balance perturbations.

Older adults exhibited ~30% less Achilles tendon stiffness than younger adults, consistent in magnitude with many previous reports on passive and active isolated contractions [16, 17]. We obtained $k_{AT}$ values during passive rotations, which explains relative differences in the magnitude of our values compared to previous literature. We opted for passive rotations given

the nature of sensing unexpected disturbances to ankle joint position. Nevertheless, this methodological decision also suggests that age-related decreases in tendon stiffness occur across a broad range of the stress-strain curve. Previous authors have reported no significant correlation between muscle strength and tendon stiffness for structures spanning the ankle [54] or knee [55] in sedentary or low-activity level young adults. Conversely, in trained cyclists and healthy young adults, respectively, Morrison et al. [56] and Muraoka et al. [57] found a significant correlation between plantarflexor strength and $k_{AT}$. Epro et al. [25] similarly found $k_{AT}$ was significantly greater in a stronger group of older women. Notably, these prior $k_{AT}$ values were taken from $MVIC_{PF}$ in contrast to the $k_{AT}$ from passive rotations reported here.

Authors with similar motivations to link MTU integrity to balance recovery have reported similar relationships in their work. For example, Epro et al. [25] differentiated two groups of older women via ankle plantarflexor strength (i.e., "strong" and "weak"). Those authors found that, following unexpected treadmill belt accelerations perturbations, stronger older women had higher $k_{AT}$ and greater $MoS_{Ant}$ than weaker older women in subsequent recovery steps. By contrast, Debelle et al. [8] found typical age-related differences in $k_{AT}$ but no age-related differences in $MoS_{Ant}$ during steady-state walking. Due to the nature of treadmill-induced slip perturbations, an anterior balance disturbance would intuitively require compensatory actions from the body in the sagittal plane. Ultimately, our data indicate similar balance responses to treadmill-induced slip perturbations in the direction of action across differences in age, muscle strength, and $k_{AT}$. However, our data reveals plantarflexor strength affects older adult's vulnerability to treadmill-induced slip perturbations in the transverse and frontal planes. Recent findings from Leestma et al. [29] using multi-directional, ground translation perturbations found that increases in WBAM following perturbations were more strongly correlated with mediolateral balance recovery metrics like step width than with anteroposterior balance recovery metrics (i.e. step width). These results indicate that walking balance and balance recovery following perturbations in the AP direction are dependent upon ML recovery strategies. Thus, while literature has focused largely on AP-direction recovery strategies during perturbed walking, future study is warranted to examine the neuromechanical responses of leg muscles acting primarily in the frontal and transverse planes following AP walking balance perturbations.

We hypothesized that higher $k_{AT}$ would be associated with better stability following treadmill-induced slip perturbations. We instead found that older adults with stiffer Achilles tendons had greater $WBAM_{TransRange}$ in response to treadmill-induced slips. As plantarflexor strength and $k_{AT}$ were not correlated in our study, we cannot attribute the increase in $WBAM_{TransRange}$ to be another consequence of faster walking speeds. Intuitively, a stiffer Achilles tendon might increase the strut-like function of the ankle by increasing force transmission from the plantarflexors at the perturbation, acting to vault the body forward. However, this affect would presumably also be apparent in $WBAM_{SagRange}$ or $\Delta WBAM_{Sag}$, where we saw no correlation with $k_{AT}$. Thus, considering this and previous work exploring similar motivations, the relationship between tendon stiffness and perturbed walking balance control is still unclear. As balance control following a perturbation is a multi-faceted neuromuscular process, it may be that tendon stiffness at a single joint is not sufficient to explain whole-body mechanics. However, due to the well-established declines in proprioceptive acuity among older adults, future investigations may benefit from examining the effect of tendon stiffness on local muscle reflex responses and balance perturbation detection.

The results of this study may be immediately applicable to clinicians seeking to mitigate falls and fall injury severity. Notably, Debelle et al. [8] concluded that training would likely have minimal effects on balance recovery after finding correlations between neither $k_{AT}$ nor plantarflexor strength and $MoS_{Ant}$. Our data corroborate this suggestion pertaining to balance recovery in the AP direction, but we provide novel insight into the importance of plantarflexor

strength for balance recovery in the frontal and transverse planes in older adults. These findings seem especially pertinent as older adults' anteroposterior stability is largely regulated by pendular mechanics, while frontal stability is largely regulated through muscular recruitment and is disproportionately afflicted during normal walking and following balance perturbations in older adults [31, 32].

This study has several experimental limitations. First, we examined WBAM in a model that neglected the arm segments and thus the specific magnitude of our outcomes and associations may differ modestly with their inclusion. Another possible limitation is that participants completed all walking trials at their preferred speed, which yielded a slower walking speed for older than younger adults. However, we believe this improves the ecological validity of our findings. Finally, we used only one perturbation paradigm (treadmill-induced slips) and treadmill-induced slip perturbation responses are likely to vary significantly between various contexts. Our study focused only on measures of MTU integrity and their relationship to gait stability following balance perturbation. However, there are many other sensory feedback mechanisms that are equally or even more involved in regulating walking balance (e.g. vestibular, visual). Our measures of Achilles tendon elongation assume a straight-line distance between the MTJ and calcaneal insertion. While there is curvature of the Achilles tendon, it primarily affects tendon force-length curves at larger values of plantarflexion, which our ankle rotation data purposefully excludes. In conclusion, we have presented novel findings indicating a potentially modifiable factor relevant to accommodating treadmill-induced slip perturbations. Plantarflexor strength can be improved through targeted training in older adults and may provide vital protection against fall risk and helping mitigate the severity of fall-related injuries. Future research should seek to identify if other muscles with mechanical actions specific to the frontal plane are also related to walking balance recovery, and if older adults have improved balance recovery following training protocols that improve plantarflexor strength. Additionally, investigating the ability of stiffer tendons to impact local muscle neural responses to balance perturbations may provide mechanistic insights into the relationship between tendon stiffness and balance control.

## Supporting information

**S1 Appendix. Plantarflexor strength and Achilles tendon stiffness relations.** Individual average peak plantarflexor torque and Achilles tendon stiffness ($k_{AT}$) for older (solid line, $n = 21$) and younger (dashed line, $n = 22$) adults. Lines of best fit for both groups are shown and significant correlations are depicted per group by their corresponding r-value accompanied by an asterisk. (p = 0.05).
(TIF)

## Acknowledgments

We would like to acknowledge Yujin Kwon for their assistance in data processing.

## Author Contributions

**Conceptualization:** Andrew D. Shelton, Gregory S. Sawicki, Jason R. Franz.

**Data curation:** Ross E. Smith, Andrew D. Shelton.

**Formal analysis:** Ross E. Smith, Andrew D. Shelton, Gregory S. Sawicki, Jason R. Franz.

**Funding acquisition:** Andrew D. Shelton, Jason R. Franz.

**Investigation:** Ross E. Smith, Andrew D. Shelton, Jason R. Franz.

**Methodology:** Andrew D. Shelton, Gregory S. Sawicki, Jason R. Franz.

**Project administration:** Gregory S. Sawicki.

**Resources:** Jason R. Franz.

**Supervision:** Gregory S. Sawicki, Jason R. Franz.

**Visualization:** Ross E. Smith, Andrew D. Shelton, Gregory S. Sawicki.

**Writing – original draft:** Ross E. Smith, Andrew D. Shelton, Jason R. Franz.

**Writing – review & editing:** Ross E. Smith, Andrew D. Shelton, Gregory S. Sawicki, Jason R. Franz.

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
