## [Decision Letter · Decision Letter 0]

2 Jan 2024

PONE-D-23-30928The effects of plantarflexor weakness and reduced tendon stiffness with aging on gait stabilityPLOS ONE

Dear Dr. Franz,

Thank you for submitting your manuscript to PLOS ONE. After careful consideration, we feel that it has merit but does not fully meet PLOS ONE’s publication criteria as it currently stands. Therefore, we invite you to submit a revised version of the manuscript that addresses the points raised during the review process. I request that you address the comments raised by both reviewers, in particular adding to/editing the presented literature where identified. A second major area which needs to be addressed are some of the issues raised related to quantification of tendon stiffness during passive rotation. I appreciate addressing this in full is ultimately a different study, but please acknowledge some of the potential measurement issues with this technique as used here.

We look forward to receiving your revised manuscript.

Kind regards,

Laura-Anne Marie Furlong

Academic Editor

PLOS ONE

“We would like to acknowledge Yujin Kwon for their assistance in data processing. This study was supported by grants from the National Institutes of Health (R21AG067388, R01AG058615).”

“This project was supported by NIH Grants R01AG058615 to JRF and GSS and R21AG067388 to JRF.

National Institutes of Health (nih.gov). These sponsors played no role in study design, data collection or analysis, decision to publish, or preparation of the manuscript.”

Reviewers' comments:

Reviewer's Responses to Questions

**Comments to the Author**

1. Is the manuscript technically sound, and do the data support the conclusions?

Reviewer #1: Yes

Reviewer #2: Partly

2. Has the statistical analysis been performed appropriately and rigorously? 

Reviewer #1: Yes

Reviewer #2: Yes

3. Have the authors made all data underlying the findings in their manuscript fully available?

Reviewer #1: Yes

Reviewer #2: Yes

4. Is the manuscript presented in an intelligible fashion and written in standard English?

Reviewer #1: Yes

Reviewer #2: Yes

5. Review Comments to the Author

Reviewer #1: This manuscript describes an interesting study into the relationships between plantarflexor strength (a modifiable risk factor) and Achilles tendon stiffnesson gait stability in an younger and aged populations. Overall the manuscript is very well constructed, I have a few suggestions listed below.

1) I would recommend introducing the current state of knowledge on Achilles tendon stiffness (and perhaps changes with age) in the introduction.

2) In the methods section (around line 102) it may be beneficial to include a range of ages instead of just a mean/stdev given the potential variability in an older adult cohort.

3) Were any repeatability/reliability studies conducted for the process of the ultrasound image collections and stiffness calculations? Both between investigators and between sessions? If so, please include information on the reliability of this approach.

4) Having seen the data figures, I am curious if your team has performed any assessments on the variability between subjects in each group? Are the data points for each perturbed step within an individual as variable as the data between subjects? Or are there any metrics that can inform the variability (sex, age, walking speed, strength, reported activity level, body size...)?

5) Were any analyses performed comparing male and female data? I see that the subject pool was divided evenly so I'm wondering if these groups were treated separately or together in statistical comparisons.

Reviewer #2: PONE-D-23-30928

The effects of plantarflexor weakness and reduced tendon stiffness with aging on gait stability

General comments

The present study aimed to examine whether there is any association between triceps surae muscle (TS) strength as well as Achilles tendon (AT) stiffness and balance recovery responses following a slip-like perturbation in older and younger adults. The authors analysed the joint moments from maximal voluntary isometric plantarflexion contractions and the elongation of the GM MTJ during passive ankle joint rotation using an isokinetic device and simultaneous ultrasonography. AP and ML margin of stability as well as whole-body angular momentum were determined post slip-like perturbations on an instrumented treadmill using optical motion capture system. The main outcomes were that older compared to younger adults had no differences in muscle strength but revealed lower passive tendon stiffness. Among the older adults, lower muscle strength was associated with greater whole-body angular momentum following slip-like perturbations. Although the overall topic is interesting for the scientific community this manuscript is in parts weak, with several methodological limitations. Moreover, the overall topic of effects of TS MTU on balance recovery responses following a perturbation has been addressed already by several previous publications and therefore does perhaps not provide significant innovation in the current form. Below I have provided detailed feedback and listed my concerns related to the current manuscript:

Specific Comments:

1. The results concerning the lack of age-related effects on TS muscle strength are not in accordance with most of the literature. In particular the reported average maximal joint moment values in younger adults seem to be quite low (less than 1 Nm/kg) and may be attributed to a lack of maximal muscle activation in the younger adults and / or to a potential misalignment between joint axis of rotation and rotational axis of the dynamometer during MVC which was not taken into account in the current study (see Arampatzis et al. Differences between measured and resultant joint moments during isometric contractions at the ankle joint. J Biomech 2005 38(4):885-92).

2. The overall topic of effects of TS MTU on balance recovery responses post perturbations has been addressed already by several previous publications e.g.

a. Debelle et al. Role of Knee and Ankle Extensors' Muscle-Tendon Properties in Dynamic Balance Recovery from a Simulated Slip. Sensors (Basel). 2022 3;22(9):3483;

b. Epro et al. Retention of gait stability improvements over 1.5 years in older adults: effects of perturbation exposure and triceps surae neuromuscular exercise. J Neurophysiol. 2018 1;119(6):2229-2240;

c. Epro et al. Effects of triceps surae muscle strength and tendon stiffness on the reactive dynamic stability and adaptability of older female adults during perturbed walking. J Appl Physiol (1985). 2018 Jun 1;124(6):1541-1549;

d. Karamanidis et al. Age-related deficit in dynamic stability control after forward falls is affected by muscle strength and tendon stiffness. J Electromyogr Kinesiol. 2008 Dec;18(6):980-9;

e. Karamanidis and Arampatzis, Age-related degeneration in leg-extensor muscle-tendon units decreases recovery performance after a forward fall: compensation with running experience. Eur J Appl Physiol. 2007 99(1):73-85;

f. Onambele et al. Calf muscle-tendon properties and postural balance in old age. J Appl Physiol 2006 100(6):2048-56;

g. Debelle et al. Recovery From a Forward Falling Slip: Measurement of Dynamic Stability and Strength Requirements Using a Split-Belt Instrumented Treadmill. Front Sports Act Living. 2020 21;2:82.

3. Most of the above-mentioned manuscripts have not been included and referred to in the introduction which clearly weakens the current section.

4. Accordingly, several statements in the introduction section such as in line 64 “However, relations between age-related decreases in tendon stiffness and walking balance control are not well-detailed in literature” or line 74 “Clearly, the literature elucidating the relationship between muscle strength and balance recovery following a perturbation is generally focused on the entire lower limb” are not correct and require amendments.

5. Page 12, hypothesis #B: “Lesser Achilles tendon stiffness (kAT) in older adults delays the transmission between ΔMTU length and muscle stretch (i.e., ΔLm), thereby delaying afferent detection of the perturbation compared to younger adults.” The provided hypothesis cannot be addressed by the current experimental design. Moreover, I argue that other sensory feedback information are perhaps more relevant to recover balance and rapidly execute effective postural corrections following a perturbation e.g. the vestibular or the visual system.

6. Line 1154: “which were interpolated over the gait cycle.” I am not sure why the elongation of the tendon over the AT force assessed during passive angle joint angular rotation on an isokinetic device was interpolated over the gait cycle.

7. The assessment of AT stiffness during passive ankle joint rotation has several limitations. For instance, during passive joint moment measurements, the measured joint moments correspond to all structures surrounding the ankle joint including muscles, ligaments, and other tendons. This is perhaps relevant at more dorsiflexed joint angle configurations potentially affecting the calculated AT force. Given that the joint moments during passive rotation were approximately less than 10 Nm (considering an AT lever arm of around 4 cm) this is a significant limitation and cannot be ignored.

8. The length change of the AT during passive ankle joint rotation as ssessed by assuming a straight line (distance between GM MTJ and AT insertion to the calcaneus) neglecting any effects of AT curvature on tendon length changes. This can significantly affect the calculated force-length relationship of the tendon.

9. Tendon resting length as well as AT lever arm affects tendon elongation during passive joint angular rotation and hence could potentially affect subject group comparison and correlation analysis. Both variables were not considered in the current approach.

10. The authors often state that they have assessed balance responses following gait perturbations. To provide more clarity, I suggest changing this to slip-like perturbations throughout the manuscript.

11. Several statements concerning previous findings and citations are wrong e.g.

a. Page 21: Mademli and Arampatzis did not assess MoS in the ML direction during walking (they determined AP MOS);

b. Page 22: Epro et al. did not assess slip-like perturbations (the authors analysed tripping).

c. Page 22: In addition, the statement “Those authors found that, following slip-like balance perturbations, stronger older women had higher kAT but did not differ in MoSAnt from weaker older women.” is not correct as weaker adults had a lower MoS during the recovery steps following first trip perturbation.

12. Description of figure 3 and 5: “… mediolateral (MoSLat) and anteroposterior (MoSLat) direction margins of stability….” and “relative mediolateral (ΔMoSLat) and anteroposterior (ΔMoSLat) direction margins of stability…”. Anteroposterior MoS is referred as MOSLat hence not correct.

6. PLOS authors have the option to publish the peer review history of their article (what does this mean?). If published, this will include your full peer review and any attached files.

Reviewer #1: No

Reviewer #2: No

---

## [Author Response · Author response to Decision Letter 0]

1 Mar 2024

Response to Reviewers for PONE-D-23-30928

“The effects of plantarflexor weakness and reduced tendon stiffness with aging on gait stability”

Reviewer #1:

We appreciate the reviewer’s thoughtful and constructive recommendations. We have incorporated each in the revised version of our manuscript as outlined in our responses that follow. 

1) I would recommend introducing the current state of knowledge on Achilles tendon stiffness (and perhaps changes with age) in the introduction.

We thank the reviewer for this helpful suggestion. Much of the current state of knowledge on Achilles tendon stiffness focuses on its effect on metabolic walking cost or its relationship to sport performance. Given the scope of our study, we took this recommendation to focus on including additional references regarding tendon stiffness the changes thereof due to age (e.g., lines 62-66) . For example: “Also, older adults are known to have age-related reductions in series elastic tissue stiffness [1-4]. In addition to disrupting metabolic walking cost and calf muscle neuromechanics [5], decreased tissue stiffness is associated with reducing maximal joint torque by increasing muscle shortening velocities [6-8] and increasing electromechanical delay [9], both of which likely impose further penalties to balance recovery in older adults.” 

2) In the methods section (around line 102) it may be beneficial to include a range of ages instead of just a mean/stdev given the potential variability in an older adult cohort.

This is another helpful inclusion, in addition to the descriptive statistics, we have also now added “age range: 65-87” to line 121.

3) Were any repeatability/reliability studies conducted for the process of the ultrasound image collections and stiffness calculations? Both between investigators and between sessions? If so, please include information on the reliability of this approach.

The specific methodological approach and outcome measures we used in this study have been previously validated in the literature. These include those covering ultrasound image collection in younger and older adults across a range of tendon force magnitudes (e.g., doi: 10.1111/j.1469-7580.2011.01365.x, doi: 10.1002/jcsm.12210) and those covering the measurement of tendon stiffnesses by combining ultrasound and dynamometry (e.g., doi: 10.1016/j.ultrasmedbio.2021.01.002, doi: 10.1519/JSC.0b013e31825bb47f). 

We performed all of our ultrasound measurements in a single experimental session, avoiding the potential for errors due to poor between-session reliability. We also did not design our study to include any repeatability/reliability measurements between investigators. Rather, our design ensured that only one investigator collected the ultrasound images. We did have two investigators identify muscle-tendon junction (MTJ) positions from the recorded ultrasound images. Compared to other anatomical features, such as a fascicle length and pennation, the position of the MTJ is very highly recognizable. In addition, to mitigate the potential influence of this methodological decision, we randomly and evenly divided participants from each age group to each of the two investigators assigned to track the MTJ positions. Although we cannot report on inter-rater reliability, we are extremely confident in the rigor of these measurements. Nevertheless, we will consider including reliability measures in our future work and appreciate the recommendation.

4) Having seen the data figures, I am curious if your team has performed any assessments on the variability between subjects in each group? Are the data points for each perturbed step within an individual as variable as the data between subjects? Or are there any metrics that can inform the variability (sex, age, walking speed, strength, reported activity level, body size...)?

We have not performed any formal analyses subject-to-subject variability. We do commend this interesting comment though and may pursue future investigations into covariates (such as those listed) to explain variability in perturbed gait balance. However, analyzing these differences in response are outside the scope of this investigation. As to the variability within subjects, the first perturbation has a larger response due to its novelty, and subsequent deliveries of the perturbation elicit lesser responses, generally.

5) Were any analyses performed comparing male and female data? I see that the subject pool was divided evenly so I'm wondering if these groups were treated separately or together in statistical comparisons.

The subject pool was indeed divided evenly in accordance with SaGER guidelines. We have not yet performed any sex-specific analyses and have thus far been exclusively focused on group-average differences between younger and older adults. For this reason, we also opted not to include those exploratory results in the revised version of this manuscript. However, based on the reviewers interesting idea, we intend to pursue an exploratory post-hoc analysis of those differences as a follow-up and will use any identified trends to pursue additional studies designed with the power to identify the influence of sex on these associations.

Reviewer #2:

We appreciate the reviewers’ attention to detail and highly productive comments and recommendations, which we believe have greatly strengthened our revised manuscript in our effort to make a meaningful contribution to the literature. We have adopted all of the reviewer’s suggestions as outlined in our responses that follow. 

1. The results concerning the lack of age-related effects on TS muscle strength are not in accordance with most of the literature. In particular the reported average maximal joint moment values in younger adults seem to be quite low (less than 1 Nm/kg) and may be attributed to a lack of maximal muscle activation in the younger adults and / or to a potential misalignment between joint axis of rotation and rotational axis of the dynamometer during MVC which was not taken into account in the current study (see Arampatzis et al. Differences between measured and resultant joint moments during isometric contractions at the ankle joint. J Biomech 2005 38(4):885-92).

We fully agree with the reviewer and appreciate the advice on potential explanations for the lack of age-related differences in TA muscle strength. We first took the opportunity to revisit our raw data to confirm the accuracy of our experimental data, to include reviewing the analog data files, mV to Nm conversions, and analysis code. We did note a minor error in our mV to Nm conversions, wherein we were estimating TS muscle strength ~12% different than actual values. In doing so, we also noted a loss of significance for one of the lesser correlations; namely, the one between PF strength and frontal plane WBAM. PF torque values have been updated throughout, to include Figures 2-6 and the appendix, the results on lines 219-220, and all instances in our revised discussion on lines 303-308, 347, 355, and 377-381. We also note here in response to the reviewer a bit more about our experimental approach. Specifically, the rotational axis of the dynamometer was well-matched to the rotation axis of each participant’s ankle, and all participants were given the same verbal instructions for the MVIC trials (i.e., “push the ball of your foot like a gas pedal as hard as possible using only your calf muscles” with loud motivational vocal cues (i.e., “push, push, push”) provided in a consistent manner to all participants. These details are included in our revised methods section on lines 146-147, which reads: “All participants received identical verbal encouragement to give maximal effort (i.e. “push the ball of your foot like a gas pedal as hard as possible using only your calf muscles”)”. Nevertheless, given the lack of an age-related difference in strength, we cannot exclude the possibility that younger adults failed to maximally activate their plantarflexors during these trials. Thus, we have added a note to address this possibility in the revised discussion section on lines 289-295, which now reads: “Another explanation may be that normalized ankle torque values among our younger adults are low when compared with previously reported values [3, 10, 11], which could result from misalignment of ankle joint rotation axes with that of the dynamometer or insufficient recruitment of ankle musculature in younger adults. As we accounted for alignment of dynamometer and ankle joint axes of rotation in our setup [12] and gave consistent vocal encouragement for maximal effort to all participants [13], it is unclear why MVIC torque values among our younger adults were low in comparison with previous values.”

2. The overall topic of effects of TS MTU on balance recovery responses post perturbations has been addressed already by several previous publications e.g.

a. Debelle et al. Role of Knee and Ankle Extensors' Muscle-Tendon Properties in Dynamic Balance Recovery from a Simulated Slip. Sensors (Basel). 2022 3;22(9):3483;

b. Epro et al. Retention of gait stability improvements over 1.5 years in older adults: effects of perturbation exposure and triceps surae neuromuscular exercise. J Neurophysiol. 2018 1;119(6):2229-2240;

c. Epro et al. Effects of triceps surae muscle strength and tendon stiffness on the reactive dynamic stability and adaptability of older female adults during perturbed walking. J Appl Physiol (1985). 2018 Jun 1;124(6):1541-1549;

d. Karamanidis et al. Age-related deficit in dynamic stability control after forward falls is affected by muscle strength and tendon stiffness. J Electromyogr Kinesiol. 2008 Dec;18(6):980-9;

e. Karamanidis and Arampatzis, Age-related degeneration in leg-extensor muscle-tendon units decreases recovery performance after a forward fall: compensation with running experience. Eur J Appl Physiol. 2007 99(1):73-85;

f. Onambele et al. Calf muscle-tendon properties and postural balance in old age. J Appl Physiol 2006 100(6):2048-56;

g. Debelle et al. Recovery From a Forward Falling Slip: Measurement of Dynamic Stability and Strength Requirements Using a Split-Belt Instrumented Treadmill. Front Sports Act Living. 2020 21;2:82.

3. Most of the above-mentioned manuscripts have not been included and referred to in the introduction which clearly weakens the current section.

We agree with this a fair and insightful comment. We especially appreciate the reviewer’s generosity in recommending several articles, one of which was unknown to our team. We originally omitted many of these articles in our introduction to provide a background more focused on the theoretical contributions of ankle muscle-tendon integrity to balance recovery, in favor of having incorporated more empirical results from these studies into the discussion. However, upon reading the introduction again with the benefit of the reviewer’s comments, we agreed that incorporating those relevant manuscripts would strengthen the revised introduction – especially in the context of other revisions we’ve adopted throughout. Thus, we have edited the introduction (lines 55-85) to more completely summarize the existing understanding of how muscle strength and tendon stiffness contribute to balance recovery, which now reads, in full for the reviewer’s convenience: 

“Balance perturbations precipitating falls are most commonly experienced during walking, where mitigation of instability requires rapid repositioning of the base of support and sufficient ground reaction forces from the musculoskeletal system to arrest momentum. In younger adults, dynamic balance recovery following gait perturbations are partly dependent on large, rapid leg extensor moments [7, 8]. With advanced age, adults experience sarcopenia [9, 10] and decreased muscle strength [11, 12], [13, 14]which would intuitively hinder balance recovery in older adults. Indeed, older adults with reduced maximal isometric leg press strength are more likely to fall [15]. Also, older adults are known to have age-related reductions in series elastic tissue stiffness [13, 14, 16, 17]. In addition to disrupting metabolic walking cost and calf muscle neuromechanics [18], decreased tissue stiffness is [13, 14, 16, 17] associated with reducing maximal joint torque by increasing muscle shortening velocities [19-21] and increasing electromechanical delay [22], both of which likely impose further penalties to balance recovery in older adults. [15]

 The relationship between age-related declines in muscle-tendon unit (MTU) integrity and dynamic stability has been explored during a variety of tasks. Due to the nature of common, real-world balance perturbations (e.g. tripping, slipping) and the ankle’s proximity to the onset of such perturbations, the triceps surae and Achilles tendon are often the locus of investigation for this relationship. Onambele et al. [14] found older adults were more unstable, had weaker ankle plantaflexors and less stiff Achilles tendons compared to younger adults showed during single-leg stance and tandem stance. During simulated forward falling, older adults took more steps to recover than younger adults, and had weaker knee and ankle extensors, as well as less stiff patellar tendons with no difference in Achilles tendon stiffness [23, 24]. While neither of these experimental tasks involve gait, both indicate age-related declines muscle-tendon integrity are important to postural stability [24-26]. Considering gait stability in response to perturbations, Epro et al. [27] showed older adult women with stronger ankle plantarflexors and stiffer Achilles tendons had better gait stability after unexpected treadmill gait perturbations. However, they later found increases in muscle-tendon capacity at the ankle through exercise had minimal effects on balance recovery in a similar population [28]. Most recently, Debelle et al. [8] showed that age related declines in muscle-tendon integrity were not related to balance recovery, as older and younger adults showed similar MoSAnt¬. This finding is unusual, as older adults are often reported to be more unstable in the AP direction than younger adults immediately following AP balance perturbations [24-26], which the authors attribute to a comparatively young “older adult” sample.” 

 We now emphasize the existing literature and more explicitly describe how it motivates the need for our study. Indeed, we continue to believe that there is a significant need for additional study in this area, specifically in regards to how older adults attenuate balance challenges applied during walking – which can be fundamentally different than postural responses during standing or simulated falling, particularly regarding mediolateral stability as evidenced by Leestma et al (https://doi.org/10.1242/jeb.244760). We specifically point this out in our revised introduction as well, stating in lines 86-92: 

“Furthermore, recent literature demonstrated that AP gait perturbations were more closely correlated with compensatory efforts in the mediolateral direction during balance recovery [14]. Given that mediolateral (ML) balance is disproportionately worsened with age [15, 16], related to increased falls risk [17], and is known to require greater active sensorimotor control from leg muscles [18], quantification of the relationship between MTU integrity and both AP and ML stability may provide novel insight into why older adults are more vulnerable to falls following gait perturbations.”. 

4. Accordingly, several statements in the introduction section such as in line 64 “However, relations between age-related decreases in tendon stiffness and walking balance control are not well-detailed in literature” or line 74 “Clearly, the literature elucidating the relationship between muscle strength and balance recovery following a perturbation is generally focused on the entire lower limb” are not correct and require amendments.

We of course also agree with this comment – as we had, for example, originally omitted our intended contrast was to highlight need for additional study in walking as opposed to standing or simulated falling. 

In addition to revising our introduction to include the relevant literature suggested, we have amended these statements to more properly acknowledge the existing literature and better define the need for and novelty of our study in lines 93-99:

“Overall, the evidence concerning age-related declines in MTU integrity and their effect on older adults’ gait stability is not clear, as older adult groups have inconsistently reported age-range cutoffs; the existing literature differ in perturbation paradigms (e.g. forward falling, mechanical pulling or “trip”, and slip); of these studies concerned with gait perturbations, none report metrics concerning mediolateral stability in either young or older adults; and Achilles and patellar tendon stiffness are inconsistently different between older and younger adults when correlated with any of the various stability metrics.”

5. Page 12, hypothesis #B: “Lesser Achilles tendon stiffness (kAT) in older adults delays the transmission between ΔMTU length and muscle stretch (i.e., ΔLm), thereby delaying afferent detection of the perturbation compared to younger adults.” The provided hypothesis cannot be addressed by the current experimental design. Moreover, I argue that other sensory feedback information are perhaps more relevant to recover balance and rapidly execute effective postural corrections following a perturbation e.g. the vestibular or the visual system.

We fully agree with the reviewer that the current experiment is not to designed to fully test the mechanistic origins of the theoretic premise we outline to describe the ways in which lesser Achilles tendon stiffness can influence the detection of muscle stretch as means of influencing vulnerability to perturbations. Fortunately, we did not design our study to test this statement itself and we are careful not to describe this theoretical premise as a specific hypotheses in our manuscript. We carefully reviewed the manuscript in full to make sure our phrasing was not in any way misleading for the reader. That said, we certainly do not disagree with the reviewer that other sensor feedback information may be perhaps more relevant to recover balance. Indeed, we have other paradigms in the lab to test, for example, the role of the visual system in controlling balance correcting responses. Thus, we sought to better acknowledge this other sensory feedback information to better place the focus of our study in the broader context of how sensory information is used to recover balance. Specifically, our revised discussion section (lines 388-391) now includes: “Our study focused only on measures of MTU integrity and their relationship to gait stability following balance perturbation. However, there are many other sensory feedback mechanisms that are equally or even more involved in regulating walking balance (e.g. vestibular, visual).”

6. Line 1154: “which were interpolated over the gait cycle.” I am not sure why the elongation of the tendon over the AT force assessed during passive angle joint angular rotation on an isokinetic device was interpolated over the gait cycle.

We thank the reviewer for identifying this typo. We have corrected this sentence on line 170 to read: “then interpolated MTJ position over the range of ankle joint rotation.”

7. The assessment of AT stiffness during passive ankle joint rotation has several limitations. For instance, during passive joint moment measurements, the measured joint moments correspond to all structures surrounding the ankle joint including muscles, ligaments, and other tendons. This is perhaps relevant at more dorsiflexed joint angle configurations potentially affecting the calculated AT force. Given that the joint moments during passive rotation were approximately less than 10 Nm (considering an AT lever arm of around 4 cm) this is a significant limitation and cannot be ignored.

We have carefully revised our manuscript to ensure that readers fully understand and appreciate the methodological limitations of our study. As we describe in our next response, we purposefully designed our stiffness calculations to involve a dorsiflexion range of motion. We also opted for passive joint rotations in particular due to, in our past experience, the potential for heel-rise during isometric contractions of higher magnitude to preclude accurate measurements of Achilles tendon stiffness. However, we agree with the reviewer that the dynamometer torque measurements during this task are in aggregate not only of the Achilles tendon but also other periarticular structures. We have revised our discussion section to disclose this potential source of error. We also fully appreciate the complex behavior of the Achilles tendon moment arm, not only as a function of ankle joint rotation, but also as a function of muscle loading. Indeed, we have published several studies characterizing these effects across multiple subject cohorts (e.g., Rasske et al. 2017, Rasske and Franz 2018, Franz et al. 2019). Thus, even in past studies using fixed-end (isometric) contractions and higher force magnitudes to estimate Achilles tendon stiffness, these complex variations in Achilles tendon moment arm are unaccounted for. Fortunately, our previously published data in Franz et al. (doi: 10.1007/s10439-018-02162-4) demonstrate that while the AT moment arm varies with ankle joint posture – these variations are dominated by changes in plantarflexion. Indeed, the moment arm is relatively invariant with posture and insensitive to the effects of muscle loading in dorsiflexion. Thus, we believe our estimates of Achilles tendon force and thus stiffness are unlikely to be affected by our decision to assume a constant moment arm. Nevertheless, we have revised our discussion of limitations to clarify these details. Specifically, we now include on lines 176-179: “We assumed constant values for Achilles tendon moment arms, though we acknowledge that these measures can vary with joint position and muscle loading [40]. However our previously-published data suggest that those variations are disproportionately small and insensitive to the effects of muscle loading in dorsiflexion”. 

8. The length change of the AT during passive ankle joint rotation as assessed by assuming a straight line (distance between GM MTJ and AT insertion to the calcaneus) neglecting any effects of AT curvature on tendon length changes. This can significantly affect the calculated force-length relationship of the tendon.

We appreciated the opportunity to clarify our methodological approaches and assumptions. We agree that the dorsoventral curvature of the Achilles tendon can – in some ankle joint postures - preclude accurate estimates of Achilles tendon elongation using a straight-line representation. We are also aware of published techniques that allow for correction of this curvature. This curvature is disproportionately problematic in ankle joint postures in larger values of plantarflexion (i.e., ≥20°). Indeed, the straight-line representation can be sufficiently rigorous when in ankle dorsiflexion. We designed our experiment with this specifically in mind and excluded all estimates of tendon elongation at postures beyond 10° of plantarflexion. Although we believe this leads to sufficiently accurate estimates to test our hypotheses, we also want to be fully transparent for the reader. Thus, we have taken the reviewer’s recommendation and added the following to our revised discussion of limitations on line 391-395-: “Our measures of Achilles tendon elongation assume a straight-line distance between the MTJ and calcaneal insertion. While there is curvature of the Achilles tendon, it primarily affects tendon force-length curves at larger values of plantarflexion, which our ankle rotation data purposefully excludes.” 

9. Tendon resting length as well as AT lever arm affects tendon elongation during passive joint angular rotation and hence could potentially affect subject group comparison and correlation analysis. Both variables were not considered in the current approach.

We refer the reviewer to our earlier response in which we outline in detail consideration of the Achilles tendon lever/moment arm. We were also careful to consider tendon resting length in the design of our experiment, and appreciated the opportunity to clarify those details in our revised manuscript. We opted to analyze ankle ranges of motion for our estimates of Achilles tendon stiffness not only to avoid errors from the tendon’s dorsoventral curvature, but also from inter-individual variation in resting length. The ankle angle at which Achilles tendon resting length occurs, assuming a knee flexion angle similar to the one used in this study, range between 10° and 20° plantarflexion. Thus, we believe our methodological approach avoided these errors. That said, we have revised our methods section to be clearer about the motivation underlying our experimental design. Specifically, we now state in lines 179-181: “We opted to analyze primarily dorsiflexion ranges of motion for our estimates of Achilles tendon stiffness not only to avoid errors from the tendon’s dorsoventral curvature, but also from inter-individual variation in resting length.”.

10. The authors often state that they have assessed balance responses following gait perturbations. To provide more clarity, I suggest changing this to slip-like perturbations throughout the manuscript.

We have taken this recommendation and revised our perturbation descriptions throughout the manuscript to “treadmill-induced slip perturbations”.

11. Several statements concerning previous findings and citations are wrong e.g.

a. Page 21: Mademli and Arampatzis did not assess MoS in the ML direction during walking (they determined AP MOS);

b. Page 22: Epro et al. did not assess slip-like perturbations (the authors analysed tripping).

c. Page 22: In addition, the statement “Those authors found that, following slip-like balance perturbations, stronger older women had higher kAT but did not differ in MoSAnt from weaker older women.” is not correct as weaker adults had a lower MoS during the recovery steps following first trip perturbation.

We appreciate the reviewer for identifying these errors in our original manuscript and certainly apologize for the oversight. We have corrected all of these errors, and also used the opportunity to review the manuscript in full to ensure we are accurately describing the available literature. Specific revisions made in response to the reviewer’s notes here include:

a. We have corrected the subscript on line 314 to refer to anteroposterior margin of stability, consistent with the work of Mademli and Arampatzis.

b. As the reviewer notes, Epro et al. did not analyze slip-like perturbations. However, upon closer review, we noticed that those authors do not refer to their perturbations as a tripping paradigm. To keep the spirit of the author’s phrasing and include the reviewer’s comment, we suggest the term “unexpected treadmill belt accelerations”, which we have used in lieu of “slip-like perturbations" on line 79 in the introduction and 339 in the discussion.

c. Again, the reviewer is correct. After review, we believe that the source of our original description was that both groups had the same MoS at the heel strike immediately before perturbation onset. We have revised this sentence to correctly manuscript state the results of the study regarding differing MoS after perturbation onset during the recovery steps in lines 339-341: “Those authors found that, following unexpected gait perturbations, stronger older women had higher kAT and greater MoSAnt than weaker older women in subsequent recovery steps.” 

These revised discussion points – which allow us to more accurately place our research discoveries in the context of the available literature – have greatly strengthened our manuscript.

12. Description of figure 3 and 5: “… mediolateral (MoSLat) and anteroposterior (MoSLat) direction margins of stability….” and “relative mediolateral (ΔMoSLat) and anteroposterior (ΔMoSLat) direction margins of stability…”. Anteroposterior MoS is referred as MOSLat hence not correct.

We have revised both figure captions to correct these errors.

References for reviewer responses

1. Franz JR, Krupenevich RL, Gray AJ, Batsis JA, Sawicki GS. Reduced Achilles tendon stiffness in aging persists at matched activations and associates with higher metabolic cost of walking. 2023.

2. McCrum C, Leow P, Epro G, König M, Meijer K, Karamanidis K. Alterations in Leg Extensor Muscle-Tendon Unit Biomechanical Properties With Ageing and Mechanical Loading. Frontiers in Physiology. 2018;9. doi: 10.3389/fphys.2018.00150.

3. Onambele GL, Narici MV, Maganaris CN. Calf muscle-tendon properties and postural balance in old age. Journal of Applied Physiology. 2006;100(6):2048-56. Epub 20060202. doi: 10.1152/japplphysiol.01442.2005. PubMed PMID: 16455811.

4. Reeves ND. Adaptation of the tendon to mechanical usage. Journal of Musculoskeletal and Neuronal Interaction. 2006;6(2):174-80. PubMed PMID: 16849829.

5. Krupenevich RL, Beck ON, Sawicki GS, Franz JR. Reduced Achilles Tendon Stiffness Disrupts Calf Muscle Neuromechanics in Elderly Gait. Gerontology. 2022;68(3):241-51. doi: 10.1159/000516910.

6. Narici MV, Maffulli N, Maganaris CN. Ageing of human muscles and tendons. Disabil Rehabil. 2008;30(20-22):1548-54. doi: 10.1080/09638280701831058. PubMed PMID: 18608375.

7. Alcazar J, Csapo R, Ara I, Alegre LM. On the Shape of the Force-Velocity Relationship in Skeletal Muscles: The Linear, the Hyperbolic, and the Double-Hyperbolic. Frontiers in Physiology. 2019;10. doi: 10.3389/fphys.2019.00769.

8. Quinlan JI, Maganaris CN, Franchi MV, Smith K, Atherton PJ, Szewczyk NJ, et al. Muscle and Tendon Contributions to Reduced Rate of Torque Development in Healthy Older Males. The Journals of Gerontology: Series A. 2018;73(4):539-45. doi: 10.1093/gerona/glx149.

9. Nordez A, Gallot T, Catheline S, Guevel A, Cornu C, Hug F. Electromechanical delay revisited using very high frame rate ultrasound. J Appl Physiol (1985). 2009;106(6):1970-5. Epub 20090409. doi: 10.1152/japplphysiol.00221.2009. PubMed PMID: 19359617.

10. Debelle H, Maganaris CN, O’Brien TD. Role of Knee and Ankle Extensors’ Muscle-Tendon Properties in Dynamic Balance Recovery from a Simulated Slip. Sensors. 2022;22(9):3483. doi: 10.3390/s22093483.

11. Karamanidis K, Arampatzis A, Mademli L. Age-related deficit in dynamic stability control after forward falls is affected by muscle strength and tendon stiffness. J Electromyogr Kinesiol. 2008;18(6):980-9. Epub 20070618. doi: 10.1016/j.jelekin.2007.04.003. PubMed PMID: 17574441.

12. Arampatzis A, Morey-Klapsing G, Karamanidis K, DeMonte G, Stafilidis S, Bruggemann GP. Differences between measured and resultant joint moments during isometric contractions at the ankle joint. J Biomech. 2005;38(4):885-92. doi: 10.1016/j.jbiomech.2004.04.027. PubMed PMID: 15713310.

13. Tod D, Iredale F, Gill N. ???Psyching-Up??? and Muscular Force Production. Sports Medicine. 2003;33(1):47-58. doi: 10.2165/00007256-200333010-00004.

14. Leestma JK, Golyski PR, Smith CR, Sawicki GS, Young AJ. Linking whole-body angular momentum and step placement during perturbed human walking. Journal of Experimental Biology. 2023;226(6). doi: 10.1242/jeb.244760.

15. Schrager MA, Kelly VE, Price R, Ferrucci L, Shumway-Cook A. The effects of age on medio-lateral stability during normal and narrow base walking. Gait & Posture. 2008;28(3):466-71. doi: 10.1016/j.gaitpost.2008.02.009.

16. Dean JC, Alexander NB, Kuo AD. The Effect of Lateral Stabilization on Walking in Young and Old Adults. IEEE Transactions on Biomedical Engineering. 2007;54(11):1919-26. doi: 10.1109/tbme.2007.901031.

17. Hilliard MJ, Martinez KM, Janssen I, Edwards B, Mille M-L, Zhang Y, Rogers MW. Lateral Balance Factors Predict Future Falls in Community-Living Older Adults. Archives of Physical Medicine and Rehabilitation. 2008;89(9):1708-13. doi: 10.1016/j.apmr.2008.01.023.

18. Bauby CE, Kuo AD. Active control of lateral balance in human walking. J Biomech. 2000;33(11):1433-40. doi: 10.1016/s0021-9290(00)00101-9. PubMed PMID: 10940402.

---

## [Decision Letter · Decision Letter 1]

27 Mar 2024

The effects of plantarflexor weakness and reduced tendon stiffness with aging on gait stability

PONE-D-23-30928R1

Dear Dr. Franz,

We’re pleased to inform you that your manuscript has been judged scientifically suitable for publication and will be formally accepted for publication once it meets all outstanding technical requirements.

Kind regards,

Laura-Anne Marie Furlong

Academic Editor

PLOS ONE

Additional Editor Comments (optional):

Reviewers' comments:

Reviewer's Responses to Questions

**Comments to the Author**

1. If the authors have adequately addressed your comments raised in a previous round of review and you feel that this manuscript is now acceptable for publication, you may indicate that here to bypass the “Comments to the Author” section, enter your conflict of interest statement in the “Confidential to Editor” section, and submit your "Accept" recommendation.

Reviewer #1: All comments have been addressed

Reviewer #2: All comments have been addressed

2. Is the manuscript technically sound, and do the data support the conclusions?

Reviewer #1: (No Response)

Reviewer #2: Yes

3. Has the statistical analysis been performed appropriately and rigorously? 

Reviewer #1: (No Response)

Reviewer #2: Yes

4. Have the authors made all data underlying the findings in their manuscript fully available?

Reviewer #1: (No Response)

Reviewer #2: Yes

5. Is the manuscript presented in an intelligible fashion and written in standard English?

Reviewer #1: (No Response)

Reviewer #2: Yes

6. Review Comments to the Author

Reviewer #1: (No Response)

Reviewer #2: (No Response)

7. PLOS authors have the option to publish the peer review history of their article (what does this mean?). If published, this will include your full peer review and any attached files.

Reviewer #1: No

Reviewer #2: No

---

## [Editor Report · Acceptance letter]

2 Apr 2024

PONE-D-23-30928R1 

PLOS ONE

Dear Dr. Franz, 

I'm pleased to inform you that your manuscript has been deemed suitable for publication in PLOS ONE. Congratulations! Your manuscript is now being handed over to our production team.

Kind regards, 

on behalf of

Dr. Laura-Anne Marie Furlong 

Academic Editor

PLOS ONE